# ReCoSeg: Residual Guided Cross-Modal Diffusion for Efficient Brain Tumor Segmentation

**Sara Yavari**[1]                                                        syavari@depaul.edu
**Rahul Pandya**[1]                                                    rpandya2@depaul.edu
**Jacob Furst**[1]                                                        jfurst@cdm.depaul.edu
[1] *DePaul University, Chicago, IL, USA*

**Editors:** Accepted for publication at MIDL 2025

## Abstract

Precise segmentation of brain tumors from MRI scans is important in clinical practice for effective diagnosis and proper treatment planning. Diffusion models have been highly effective in image generation and segmentation. In this work, we introduce ReCoSeg, a novel semi supervised framework that combines cross-modal diffusion based synthesis with residual guided segmentation to improve accuracy. First, a diffusion model synthesizes the T1ce modality from existing FLAIR, T1, and T2 MRI scans. The synthesized and the real image difference captured as residuals—highlights potential tumor regions. Residuals are then used as attention cues in a lightweight U-Net for segmentation, reducing the reliance on dense labels with an enhanced segmentation performance.

**Keywords:** Brain Tumor Segmentation, Diffusion Models, Residual Maps, Semi-Supervised Learning.

## 1. Introduction

Brain tumor segmentation from MRI scans is a critical neuro-oncology task supporting diagnosis, treatment planning, and longitudinal monitoring. Deep learning models—specifically U-Net (Ronneberger et al., 2015) and its three-dimensional extensions: 3D U-Net (Çiçek et al., 2016)—are now standard tools for this purpose, with very high accuracy through end-to-end supervised learning paradigms(Ronneberger et al., 2015). The models depend heavily on significant quantities of annotated data and are typically computationally intensive (Luo et al., 2021). This renders them less scalable and applicable in real-world clinical settings, where annotations are limited and modality availability can vary. More recent approaches have employed generative models—namely Denoising Diffusion Probabilistic Models (DDPMs) to perform cross-modal synthesis to synthesize missing imaging modalities (Zhan et al., 2024). DDMM-Synth (Wang et al., 2023) effectively uses diffusion models to predict the T1ce modality from other MRI sequences. Our novel proposed method is ReCoSeg, which follows a two-step approach: Firstly, it uses a diffusion model to generate one MRI modality from others. The residuals are computed, which highlight tumor regions used for effective segmentation. ReCoSeg mimics how radiologists identify anomalies among different scan types by delineating the generation and detection procedures. This enhances tumor segmentation to become more accurate, readable, and efficient.

## 2. Method

The proposed ReCoSeg pipeline is a two-stage system that synergistically integrates cross-modal MRI synthesis with residual-guided tumor segmentation. This design enables the model to identify tumor regions by leveraging discrepancies between real and synthesized modalities, mimicking how radiologists interpret multi-sequence MRIs(Ellingson et al., 2015). In the first stage, we employ a Denoising Diffusion Probabilistic Model (DDPM) to synthesize the T1ce modality from the available FLAIR, T1, and T2 MRI sequences. DDPM is trained using the standard forward-reverse diffusion process (Ho et al., 2020) with T1ce as the target modality. To guide the reconstruction, we optimize the model using a weighted combination of Binary Cross-Entropy (BCE) loss and Dice loss, encouraging voxel-wise accuracy and spatial alignment with the ground truth. In the second phase, once the synthetic T1ce image is generated, we compute a residual map by calculating the absolute difference (Osher, 2013) between the predicted and actual T1ce images. This residual highlights regions which correspond to tumor areas. We concatenate the residual map with the original input modalities for the final segmentation and feed them into a lightweight 2D U-Net (Liao et al., 2024). This residual-supervised signal guides the segmentation model focuses on potential tumor regions, enhancing its ability to localize abnormalities with reduced dependence on densely annotated training data.

### 2.1. Experiments

We compare ReCoSeg with three relevant baselines to evaluate its performance. UNet2D (Ronneberger et al., 2015) is a widely used 2D U-Net trained with full supervision on all four modalities (FLAIR, T1, T2, T1ce), serving as a strong conventional benchmark. UNet3D (Çiçek et al., 2016) extends this to volumetric segmentation with 3D inputs, offering a comparison against high-capacity models using spatial context. DDMM-Synth (Wang et al., 2023) employs a diffusion model to synthesize T1ce from the other modalities, similar to our method, but without residual guidance. This selection enables fair and diverse comparison across traditional, volumetric, and generative settings. All models are trained on 2D axial slices from the BraTS2020 dataset (355 subjects), resized to $120 \times 120$ and normalized. The diffusion model is trained for 1,000 steps using Adam ($2 \times 10^{-4}$). All models use BCE + Dice loss in PyTorch 2.0 on an RTX 3090 GPU.

Table 1: Comparison with Baselines on BraTS2020

| Model | Dice | IoU |
|---|---|---|
| UNet2D | 0.784 | 0.736 |
| UNet3D | 0.842 | 0.743 |
| DDMM-Synth | 0.872 | 0.811 |
| **ReCoSeg (Ours)** | **0.917** | **0.853** |

## 2.2. Result

The novelty of ReCoSeg is in the breakdown of the segmentation task into two phases: an unsupervised generative phase and a supervised discriminative phase. This split enhances interpretability as well as performance. The residual map provides a transparent reasoning behind the model's segmentation decisions. Table 1 shows the quantitative comparison, in which ReCoSeg achieves a Dice score of 0.9167 and IoU of 0.853, outperforming fully supervised and generative baselines, while maintaining a modular and lightweight design. By restricting the diffusion model to stage 1, ReCoSeg ensures computational efficiency while producing interpretable outputs via modality-based anomaly detection. Its lightweight segmentation design remains robust with missing modalities and is suitable for resource-limited environments. Figure 1 illustrates how residual guided segmentation enhances tumor localization.

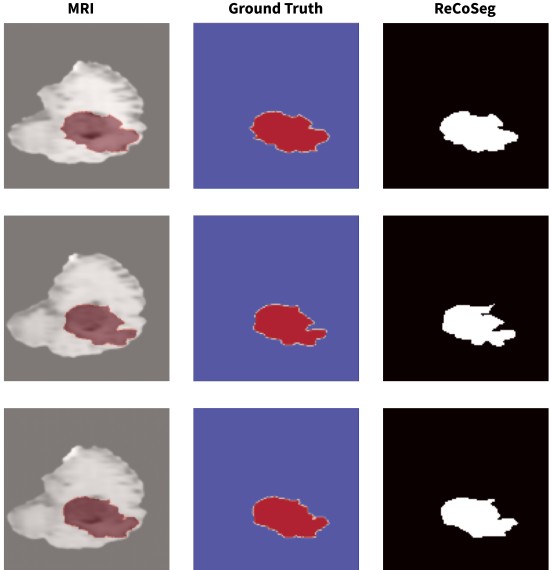

Figure 1: Segmentation results from ReCoSeg, showing (left to right): MRI (middle-slice) with tumor highlighted, ground truth, & predicted tumor segmentation.

## 3. Conclusion and Future Work

ReCoSeg, a semi-supervised method combining cross-modal diffusion and residual guided learning for binary brain tumor segmentation offers an interpretable, efficient approach aligned with radiological workflows. Residual maps serve as attention cues, improving accuracy while reducing reliance on dense annotations. The results show competitive performance even with missing or corrupted modalities, highlighting ReCoSeg's robustness for clinical use. Future work can include attempts to further increase the robustness of the technique with 3D diffusion, transformer-based context modeling, multi-scale residuals, and potential future generalization to multiclass segmentation tasks.

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
