# OpenReview forum: "ReCoSeg: Residual-Guided Cross-Modal Diffusion for Efficient Brain Tumor Segmentation"
_MIDL.io/2025/Short_Papers — MIDL 2025 - Short Papers_

### Official Review · Reviewer_HYjf · 2025-04-25

**Rating:** 3
**Confidence:** 5

**Summary:**

The authors present ReCoSeg, a semi-supervised segmentation approach that combines cross-modal diffusion synthesis with residual guidance. The method generates pseudo-T1ce images from FLAIR, T1, and T2 modalities through a diffusion model, computes pixel-wise differences between synthetic and real T1ce images to create a tumor attention map, and subsequently employs this map to guide segmentation. Experimental results demonstrate that ReCoSeg outperforms fully-supervised baseline methods on the BraTS2020 dataset.

**Strengths:**

Clear Methodology and Contribution: The two-stage pipeline—(1) diffusion-based T1ce synthesis and (2) residual-guided segmentation—is articulated with exceptional clarity and comprehensibility, presenting a coherent framework for brain tumor segmentation.

**Weaknesses:**

1. Inadequate Acknowledgment of Prior Research: The utilization of diffusion models to learn normal tissue distributions and generate anomaly attention maps has been previously established in unsupervised and weakly-supervised approaches (e.g., Pinaya et al., MICCAI 2022; Wolleb et al., MICCAI 2022). The authors fail to sufficiently contextualize their work within this existing research landscape.

2. Insufficiently Substantiated Semi-Supervised Evaluation: While claiming a semi-supervised experimental setup, the authors do not specify the distribution between labeled and unlabeled data. They report training on 355 BraTS2020 cases, yet the official training set comprises 369 annotated volumes and a validation set of 125 unlabeled volumes (as per the BraTS2020 dataset documentation).

Reference:

[1] Pinaya, W. H. L., et al. "Fast Unsupervised Brain Anomaly Detection and Segmentation with Diffusion Models." MICCAI 2022.

[2] Wolleb, J., et al. "Diffusion Models for Medical Anomaly Detection." MICCAI 2022.

[3] BraTS2020 Dataset: https://www.kaggle.com/datasets/awsaf49/brats20-dataset-training-validation/

---

### Decision · Program_Chairs · 2025-05-01

Accept